# Effects of a Three-Day vs. Six-Day Exposure to Normobaric Hypoxia on the Cardiopulmonary Function of Rats

**DOI:** 10.3390/cimb47020125

**Published:** 2025-02-14

**Authors:** Charly Bambor, Sarah Daunheimer, Coralie Raffort, Julia Koedel, Aida Salameh, Beate Raßler

**Affiliations:** 1Carl-Ludwig-Institute of Physiology, University of Leipzig, 04103 Leipzig, Germany; charly.bambor@gmx.net (C.B.); sarah.daunheimer@gmx.de (S.D.); 2Department of Pediatric Cardiology, Heart Centre, University of Leipzig, 04289 Leipzig, Germany; coralie.raffort@uni-leipzig.de (C.R.); aida.salameh@medizin.uni-leipzig.de (A.S.); 3Institute of Pathology, University of Leipzig, 04103 Leipzig, Germany; julia.koedel@medizin.uni-leipzig.de

**Keywords:** prolonged normobaric hypoxia, cardiac function, left and right ventricular catheterization, pulmonary edema, pulmonary inflammation, tumor necrosis factor α, pleural fluid

## Abstract

In rats, normobaric hypoxia significantly reduced left ventricular (LV) inotropic function while right ventricular (RV) function was not impaired. In parallel, the animals developed pulmonary edema and inflammation. In the present study, we investigated whether cardiac function and pulmonary injury would aggravate after three and six days of hypoxia exposure or whether cardiopulmonary reactions to prolonged hypoxia would become weaker due to hypoxic acclimatization. Sixty-four female rats were exposed for 72 or 144 h to normoxia. They received a low-rate infusion (0.1 mL/h) with 0.9% NaCl solution. We evaluated indicators of the general condition, blood gas parameters, and hemodynamic function of the rats. In addition, we performed histological and immunohistochemical analyses of the lung. Despite a significant increase in hemoglobin concentration, the LV function deteriorated with prolonged hypoxia. In contrast, the RV systolic pressure and contractility steadily increased by six days of hypoxia. The pulmonary edema and inflammation persisted and rather increased with prolonged hypoxia. Furthermore, elevated protein concentration in the pleural fluid indicated capillary wall stress, which may have aggravated the pulmonary edema. In conclusion, six days of hypoxia and NaCl infusion place significant stress on the cardiopulmonary system of rats, as is also reflected by the 33% of premature deaths in this rat group.

## 1. Introduction

Exposure to a hypoxic environment or pathologic conditions causing a reduced or insufficient oxygen supply is a major challenge to the organism, requiring a multitude of adaptational reactions to compensate for the oxygen deficiency. The cardiopulmonary system makes an important contribution to the hypoxia acclimatization. However, if the heart and lungs themselves are affected by the lack of oxygen, the range of compensatory responses is limited, which can further reduce the supply of oxygen to all organs and tissues.

An impaired energy metabolism is a direct consequence of oxygen deficiency. Tissues reduce their demand for oxygen when possible. Anaerobic energy production increases, but this is less efficient than aerobic energy generation and, even more importantly, is associated with the development of metabolic acidosis. The hypoxic ventilatory response (HVR) is one of the earliest compensatory reactions to hypoxia. The alveolar, and consequently, the arterial pO_2_ are improved by increased excretion of CO_2_. Hypoxic pulmonary vasoconstriction (HPV) contributes to the optimization of alveolar gas exchange. Hypoxia-induced activation of the sympathetic nervous system by oxygen-sensing mechanisms such as carotid bodies [1], and the resulting release of various stress hormones, increase cardiac output, and thus, the circulatory component of oxygen supply to the organs [2,3]. Finally, a sustainable mechanism of adaptation to hypoxia is the improvement of the oxygen transport capacity of the blood by increased erythropoiesis [4].

The adaptable and advantageous reactions of the organism, however, carry the potential for the development of complications. The elevated hematocrit means an additional load on the cardiovascular system. This is attenuated by a reduction in plasma volume [3,5,6], which in turn, may reduce end-diastolic filling of the ventricles and limit the enhancement of cardiac output. Another typical and critical pathology of hypoxia is the formation of pulmonary edema (PE). The specific form of hypoxic PE occurring at high altitude is called high-altitude pulmonary edema (HAPE). Hypoxic PE is considered to be a hydrostatic edema caused by increased pulmonary capillary pressure. It results from enhanced and uneven HPV, causing overperfusion, and thus, an increase in pulmonary capillary pressure in lung regions with weaker HPV [7]. Strong sympathetic activation or elevated plasma catecholamine levels can have pro-edematous effects even in normoxia [8,9], suggesting that sympathetic activation under hypoxic conditions may promote and aggravate the formation of PE. If the pulmonary capillary pressure strongly increases, approaching about 40 mmHg, capillary wall stress may result [10,11]. This damage to the alveolocapillary barrier may cause fluid intrusion into the alveoli, a complication that can often be fatal.

### Cardio-Circulatory and Pulmonary Reactions to Hypoxia in Humans and Animals

The human heart responds to acute hypoxia with an increase in cardiac output resulting from tachycardia accompanied by an unchanged stroke volume [12,13,14,15]. In addition, echocardiographic studies demonstrated an improved LV twist mechanics in humans at high altitude [15,16,17]. This improvement in cardiac function is supposed to result from sympathetic activation, which occurs in humans within 30 min of moderate hypoxia [18]. In addition, hypoxia induces vasodilation in the systemic circulation [19,20]. The resulting decrease in the total peripheral resistance (TPR) contributes to the enhancement of cardiac output.

In contrast, numerous animal studies have demonstrated a deterioration in left ventricular (LV) function under hypoxic conditions. It is thought to result from reduced myocardial oxygen consumption and anaerobic metabolism [21]. More specifically, mitochondrial respiration and ATP synthesis are compromised in hypoxic LV myocytes [22,23]. In addition, hypoxia impairs LV mechanical function, specifically, the twist of the apical myocardium, as was shown in pigs under acute myocardial ischemia. As a consequence, the pressure and contractility of the LV, as well as stroke volume (SV) and ejection fraction (EF), are reduced [24]. Heart rate (HR), aortic blood flow, and TPR decreased in acute hypoxia [25]. This was also confirmed in our previous studies on rats, as they showed that only a few hours of exposure to normobaric hypoxia with 10% O_2_ significantly reduced LV systolic pressure (LVSP) and contractility (LV dP/dt max). After 24 h of hypoxia, LVSP and LV dP/dt max decreased to about 80% and 65% of normoxic values, respectively [26]. Despite an only mild reduction in HR by about 10%, the cardiac output was significantly reduced to about 60% of the normoxic value [27], indicating a marked reduction in stroke volume. With longer hypoxic exposure (72 h), LV function did not recover, which may be explained by nitrosative stress and apoptosis of the myocardial cells [28]. Contrary to the LV, the maximal pressure and contractility of the right ventricle (RV) did not decrease, but rather, increased slightly over 3 days of hypoxia even though nitrosative stress and apoptosis were also present in the RV [26,28]. These hemodynamic changes were paralleled by inflammation and the formation of PE [27,29]. The PE appeared after about 6 h of hypoxia and increased steadily until 72 h of hypoxia exposure [26,27,29]. After 3–7 days of hypoxia, we observed regression of PE, blood congestion, and inflammation in the lungs of rats, while mRNA of collagen type I and III gradually increased in this period [30].

The present study was performed to investigate the cardiopulmonary changes during a prolonged but still sub-chronic exposure to hypoxia. Most animal studies on the effects of hypoxia have applied either short-term (1–3 days) or long-term (2–4 weeks) of hypoxia exposure. As we were interested in investigating an intermediate time interval, we compared the effects of three-day and six-day exposure to hypoxia on the heart and lungs of rats. We hypothesized that hypoxia acclimatization occurs and is detectable by a gradual increase in hemoglobin concentration (cHb) and hematocrit (Hct) in the blood. Consequently, we would expect that hypoxia tolerance of the rats would be improved or at least maintained. Hence, we assumed that LV and RV functions remain largely stable between the end of the third and sixth day of hypoxia. Pulmonary edema and inflammation were expected to recede during this period. Many other studies on hypoxia explored either cardiac function or pulmonary function, but there are only very few studies that investigated the effects on both systems and their possible interrelations. With its experimental approach, the present study might contribute to close this gap, thus deepening the insight into the complex systemic effects of generalized hypoxia.

## 2. Materials and Methods

### 2.1. Animal Model

The experiments were performed on 64 female Sprague Dawley rats supplied by Charles River (Sulzfeld, Germany). The body weight of the animals was 240 ± 1.9 g, corresponding to an age of about 10–12 weeks. All animal protocols were approved by the state agency (Landesdirektion Sachsen, number and date of approval: TVV 46/18; 17 December 2018). The experiments were conducted in accordance with the Guide for the Care and Use of Laboratory Animals published by the National Institutes of Health and with the “European Convention for the Protection of Vertebrate Animals used for Experimental and other Scientific Purposes” (Council of Europe No 123, Strasbourg 1986).

### 2.2. Study Protocol

All animals received an intravenous infusion with 0.9% NaCl solution at a rate of 0.1 mL h^−1^ over the total experimental time. Infusions were administered with automatic pumps (Infors AG, Basel, Switzerland) via an infusion catheter (Vygon, Aachen, Germany), which was inserted into the left jugular vein. The animals were divided at random into two cohorts to be exposed to normoxia (N) or normobaric hypoxia (H). Each cohort was subdivided into two groups. One group of each cohort remained for 72 h in normoxia (72N, *n* = 14) or hypoxia (72H, *n* = 18), and the other group was exposed to the respective conditions over 144 h (144N, *n* = 8; 144H, *n* = 24). Normoxic animals were kept under room air conditions. The animals in the hypoxic cohort were placed into a hypoxic chamber sized 65 × 105 × 50 cm. The gas mixture in the chamber contained 10% oxygen in nitrogen. Special equipment prevented penetration of ambient air during manipulations on the animals, thus keeping the oxygen concentration in the chamber stable at 10 ± 0.5%. Exposure to the hypoxic environment started immediately after insertion of the infusion catheter. Throughout the duration of the experiment, the animals were awake and moved freely, with access to tap water and rat chow diet (Altromin C100, Altromin GmbH, Lage, Germany).

### 2.3. Hemodynamic Measurements

About 40 min before the end of the exposure time, the animals were anesthetized with thiopental (Trapanal^®^ 80 mg kg 1, i.p.). When the anesthesia had reached a sufficient depth, the animals were weighed and then placed onto a heated plate to avoid cooling during the hemodynamic measurement. First, the animals were tracheotomized, and a polyethylene cannula was placed in the trachea. Then, the right ventricle (RV) was catheterized with Millar (Millar Instruments, Houston, TX, USA) ultraminiature catheter pressure transducers. For catheterization of the left ventricle (LV) and recording of pressure–volume loops, we inserted a pressure–volume Millar catheter (Millar Instruments, Houston, TX, USA) into the left ventricle via the right carotid artery. The catheter was connected to a signal amplifier system (MPVS Ultra, Millar Instruments, Houston, TX, USA). For data acquisition and analysis, we used the Power Lab 16/35 and Lab Chart Pro Software (version Lab Chart 8) from ADInstruments (sales department FMI Föhr Medical Instruments GmbH, Seeheim, Germany). Parallel conductance was corrected by injection of 0.1 mL of 0.9% NaCl solution and calibration of stroke volume by the thermodilution method. The following variables were measured in RV and LV: LVSP, RVSP, HR, LV dP/dt max and dP/dt min, and RV dP/dt max and dP/dt min as measures of ventricular contractility and relaxation, respectively. From LV catheterization, we determined SV, EF, stroke work (SW), end-diastolic pressure (edP), and end-diastolic volume (edV). After withdrawal of the LV catheter tip into the aorta, diastolic aortic pressure (DAP) was measured to calculate mean aortic pressure (MAP). Cardiac output was measured by thermodilution using a thermosensitive 1.5F microprobe and a Cardiomax II computer (Columbus Instruments, Columbus, OH, USA). From this measurement, we calculated cardiac index (CI) as body mass-related cardiac output and TPR as the quotient of MAP and CI. Hypoxic animals remained in hypoxia until completion of hemodynamic measurements.

### 2.4. Sampling of Materials

After the hemodynamic measurements, the animals were sacrificed by drawing blood from the abdominal aorta. From a small sample of aortic blood, oximetry and blood gas assessment were performed using a blood gas analyzer ABL800 BASIC (Radiometer Medical ApS, Brønshøj, Denmark). For measurement of blood glucose concentration, we used the blood glucose monitor BGStar (AgaMatrix, Inc., Salem, NH, USA). Then, the thoracic wall was opened and pleural fluid (PF) was collected. We ligated the right main bronchus and performed a bronchoalveolar lavage (BAL) of the left lung. The recovered BAL fluid was frozen and stored at −80 °C for further analyses. The left lung was discarded thereafter. The intact right lung and the heart were excised and weighed. The cardiac apex and pieces from the right lung were fixated in formalin for histological analysis. In addition, we took a piece of the middle lobe of the right lung for determination of wet-to-dry weight (W/D) ratio.

Finally, we weighed the feed residues to determine the total feed consumption of the animals. The total water uptake over the experimental time was recorded from the majority of the animals.

### 2.5. Lung Histology

The formalin-fixated tissue samples of the right lung were embedded in paraffin, sliced, and stained with hematoxylin–eosin. Two independent investigators (S.D. and J.K.), who were blinded to the treatment group, evaluated PE and congestion in the lungs. For a detailed quantification of PE, the complete histological section of a lung was assessed. First, the width of the alveolar septa and the definition of alveolar spaces were evaluated in each area of the section to determine PE severity (expressed as PE score). PE scores ranged from 0 (absent) to 1 (mild: alveolar septa slightly thickened, alveolar space well defined), 2 (moderate: thickness of alveolar septa about double the normal width, alveolar space narrowed but still defined), and 3 (severe: alveolar spaces hardly determinable and/or alveolar edema). Second, the PE index (PEI) was calculated by cumulating the products of the PE score and the proportionate area of each part of the histological preparation.

### 2.6. Immunohistochemistry

Immunohistochemistry was used to detect tumor necrosis factor (TNF) α as a marker of inflammation in the lung. Specimens from the right lung, fixed with 4% formalin, were embedded in paraffin, and 2 µm slices were cut. After mounting on microscopic slides, the samples were dewaxed and rehydrated. The dewaxed and rehydrated specimens were cooked in 0.01 M citrate buffer (pH = 6) and then blocked with bovine serum albumin (BSA) to saturate unspecific bindings. The specimens were treated with rabbit monoclonal anti-TNFα primary antibody (1:100, Sigma-Aldrich, Taufkirchen, Germany) overnight at 4 °C. Then, they were washed again in Tris-buffer, and the appropriate goat anti-rabbit secondary antibody (1:200, Sigma-Aldrich, Taufkirchen, Germany), labeled with horseradish peroxidase (HRP), was applied for 1 h. After a further washing step, a peroxidase reaction was carried out using the red chromogen 3-amino-9-ethylcarbazole (AEC, Enzo, Lörrach, Germany) according to the manufacturer’s instructions. Cell nuclei were counterstained with hemalum. All specimens were investigated microscopically using the Axioimager M1 microscope from Zeiss (Carl Zeiss, Jena, Germany). As TNFα is mainly located in the bronchial and peribronchial regions, photographs were taken from these regions using an AxioCam MRc 5 camera and Zen Blue 3.1 software (Carl Zeiss, Jena, Germany) at 20× magnification. At least 8 pictures per animal were evaluated by a blinded observer (S.D.). The program ImageJ (version 1.54m) [31] was used for measurements of the TNFα-positive area (in µm^2^) in the pictures. The expression of TNFα is given as the TNFα-positive area related to the bronchial surface area of the specimen (in percent).

### 2.7. Lung Wet-to-Dry Weight Ratio

Lung tissue samples were weighed immediately after preparation (wet weight, W) and after drying in an oven at 75 °C for 48 h (dry weight, D). The W/D ratio served as a surrogate parameter of fluid accumulation in the lung and, thus, as an indicator of pulmonary blood congestion and edema.

### 2.8. Protein Concentration in Serum, BAL Fluid, and Pleural Fluid

Total protein concentration in serum and BAL was determined using the BCA protein assay from Pierce (Thermo Fisher Scientific, Dreieich, Germany) according to the manufacturer’s instructions. For BAL fluid analysis, the undiluted supernatant of the first lavage was used. Pleural fluid (PF) and serum (S) were diluted with PBS at a ratio of 1:200. A standard curve was generated with bovine serum albumin (25 µg/mL to 2000 µg/mL) and measured together with the unknown samples at 562 nm. A two-fold determination was performed using the spectrophotometer Synergy HTX form BioTek (now Agilent, Waldbronn, Germany). The protein concentrations [P] in the fluids are given in g/L. The concentration ratio [P] PF/[P] S was calculated for each individual animal and is given in %.

### 2.9. Statistical Analysis

Statistical analyses were carried out with the software package SigmaPlot Version 14.0 (Systat Software GmbH, Erkrath, Germany) for Windows. We used Analysis of Variance (ANOVA) procedures to compare the groups for significant differences. At first, a Shapiro–Wilk test of normality was performed. In case of normal distribution, we used a One-Way ANOVA with a post hoc test according to Fisher’s LSD method. If the data were not normally distributed, a Kruskal–Wallis ANOVA on ranks with a post hoc test according to Dunn’s method was applied. Both post hoc tests are multiple comparison procedures comparing all possible pairwise mean differences. The *p* values < 0.05 were considered significant. In order to assess possible effects of LV and RV function and of hypoxia on the severity of PE under hypoxia, we performed a multiple linear regression using PEI as the dependent variable and LVSP, RVSP, and hypoxia as independent variables (predictors). The correlation coefficient and the coefficient of multiple determination of the model were interpreted according to Cohen [32].

## 3. Results

### 3.1. General Outcome

The hypoxic animals did not tolerate the experiment as well as the normoxic animals. From all the rats in normoxic conditions (72 h normoxia, 72N; 144 h normoxia, 144N), only one died prematurely from an injury to the carotid artery during LV catheterization. In the hypoxic cohort (72 h hypoxia, 72H; 144 h hypoxia, 144H), 17 rats (72H *n* = 4; 144H *n* = 13) showed signs of LV decompensation and acute right ventricular failure (aRVF) at necropsy, with the most prominent features being a huge and dilated RV; a purple, patchy lung; massive blood pooling into the inferior vena cava; and a huge, dark purple liver. More than half of these animals (72H *n* = 2; 144H *n* = 8) died in thiopental anesthesia before or at the beginning of the hemodynamic measurements. None of the normoxic animals showed signs of LV decompensation and aRVF or died with these signs.

The daily water intake of the normoxic rats was 31.0 ± 3.2 mL/d, significantly higher than the hypoxic animals (72H 14.4 ± 3.5 mL/d, *p* = 0.009; 144H 19.7 ± 2.0 mL/d, *p* = 0.036). Food intake was also significantly reduced in the hypoxic animals by about 50% compared with the normoxic rats (Figure 1). Consistently, the normoxic rats largely maintained their body weight (BW) (ΔBW −2.4 ± 1.6% of baseline). In contrast, the hypoxic rats showed a significant loss in BW by 11.7 ± 1.1% in three days (*p* < 0.001 compared with 72N) and even by 16.8 ± 0.8% in six days of hypoxia (*p* < 0.001 compared with 144N and 72H; Figure 1).

### 3.2. Blood Analysis

In hypoxic animals, the arterial oxygen saturation (SaO_2_) and partial pressure of oxygen (paO_2_) were reduced compared with normoxic rats, but the difference became significant only after six days of hypoxia (*p* < 0.001 and *p* = 0.015, respectively). Hypoxemia was accompanied by the development of acidosis. Arterial pH decreased to 7.32 after three days but dropped even further by six days of hypoxia (7.15, *p* < 0.001) compared with the related normoxic group (Figure 2). The arterial partial pressure of carbon dioxide (paCO_2_) was markedly reduced in the 72H group (33.3 mmHg) but re-increased to normal values after six days of hypoxia (40.7 mmHg, *p* > 0.05). Lactate and glucose concentrations increased slightly but not significantly in the hypoxic groups, particularly in the 144H group. Moreover, hypoxic animals showed a significant increase in K^+^ and Na^+^ concentrations after three days of hypoxia (*p* = 0.005 and 0.01, respectively), which partly receded by day six of hypoxia (Table 1). Hemoglobin concentration (cHb) and hematocrit (Hct) increased gradually until the sixth day of hypoxia (*p* < 0.001 compared with the 144N group), indicating progressive adaptation to hypoxia (Figure 3).

### 3.3. Hemodynamic Measurements and Heart Weight

The hemodynamic results are presented in Figure 4 and Table 2. As expected, LVSP was significantly reduced after three days of hypoxia (*p* = 0.001 compared with 72N). By the sixth day, it re-increased to almost normal levels (*p* = 0.22 compared with 72N). LV dP/dt max, diastolic aortic pressure (DAP), mean aortic pressure (MAP), LV end-diastolic volume (edV), SV, EF, and stroke work (SW) showed a similar trend, but the differences were mostly not significant. In contrast, LV end-diastolic pressure (edP) increased mildly in the 72H group, but increased massively to about three-fold values in the 144H group (*p* < 0.001 compared with 144N). Of note, LVSP and LV edV were also reduced in the 144N group. LV relaxation (dP/dt min) decreased slightly, but not significantly, under hypoxia. HR decreased significantly after three days and even further after six days of hypoxia (*p* < 0.05 compared with the time-corresponding normoxic group). Consequently, the cardiac index (CI) gradually decreased under hypoxic conditions and was significantly diminished in the 144H group (*p* = 0.042 compared with 144N). TPR remained stable over the first three days of hypoxia, but then increased markedly by almost 50% (*p* < 0.001 compared with 144N).

In contrast to LV, the SP and dP/dt max in the RV slightly increased under hypoxia, and this trend was even stronger after six days of hypoxia. At this time, RVSP was significantly higher than in the time-corresponding normoxic control (*p* = 0.005). Relative heart weight (HW/BW) increased slightly but not significantly in the hypoxic rats compared with the 72N rats, indicating that no significant hypertrophy had developed in the heart. Of note, heart weight (HW) was, in the 144N group, about 10% lower than in the 72N group.

### 3.4. Pulmonary Injury

Lung histology revealed the existence of a mild interstitial edema in the lungs. Even normoxic rats were not completely free from signs of PE, but PE became more severe under hypoxic conditions. The pulmonary edema index (PEI) increased over time also in normoxia, but even more in hypoxia (Figure 5). However, the PEI differences among the groups were not significant. Relative lung weight (LuW/BW), which can be considered an indicator of blood congestion and/or edema in the lungs, progressively increased with advancing duration of hypoxia and reached about 140% of normoxic values after 144 h (*p* = 0.003. Figure 6). The pulmonary wet-to-dry weight (W/D) ratios ranged between 4.9 and 5.6 without significant differences between the groups, confirming that the edema was mainly confined to the interstitium and did not enter the alveoli. During the formation of PE, fluid filtration into the pleural space can serve as a drainage route. In hypoxic animals, the amount of pleural fluid (PF) was almost twice as high as in normoxic animals, even if the inter-individual variation was large (Figure 6). The PF, which is normally low in protein, contained a relatively high amount of protein, in particular in the 72H group with more than 30 g/mL (Table 3). A protein concentration in PF > 30 g/mL or a PF-to-serum protein concentration ratio ([P] PF/[P] S) > 0.5, as achieved in the 144H group, indicates exudation, which can result from damage to the capillary walls [33,34]. In contrast, the bronchoalveolar lavage fluid (BALF) protein concentration was low and in a similar range in all groups, confirming that the edema had not entered the alveoli. Immunohistochemical analyses demonstrated that the hypoxic PE was accompanied by inflammation, which was preferably localized in the peribronchial regions. The expression of TNFα as a potent proinflammatory cytokine reached in the 144H group about twice the expression of that in the normoxic groups (*p* < 0.001 vs. 144H; Figure 7). In the 144N group, TNFα remained at the level of the 72N group.

### 3.5. Effects of Cardiac Dysfunction on Pulmonary Edema

The multiple linear regression revealed a correlation coefficient (R) of 0.438. As R = 0.30 indicates moderate correlation and R = 0.50 indicates strong correlation [32], our result can be interpreted as a moderate to strong correlation between the predictors LVSP, RVSP, and hypoxia, and PEI as the dependent variable. The coefficient of multiple determination (R^2^) is the proportion of variation in the dependent variable PEI that can be explained by the multiple regression model based on the three predictors LVSP, RVSP, and hypoxia. The R^2^ for the overall model was 0.192 (adjusted R^2^ = 0.133), indicative of a moderate goodness-of-fit according to Cohen [32]. The predictors LVSP, RVSP, and hypoxia significantly predict the criterion PEI (F(3, 41) = 3.253, *p* = 0.031). The standardized coefficients (Beta) indicate a very weak (Beta = 0.093) and weak (Beta = 0.143) positive effect of LVSP and RVSP, respectively, on PEI with *p* > 0.05, but a significant effect of hypoxia on PEI (Beta = 0.367, *p* = 0.025).

## 4. Discussion

### 4.1. Effects of Prolonged Hypoxia on the General Condition of the Animals

The present data clearly show that acclimatization to hypoxia is verified by a significant increase in Hct and cHb after three days and even more after six days of hypoxia exposure. Despite this compensatory reaction, the general condition of the hypoxic animals was poor, and it deteriorated with longer exposure to hypoxia. The observed body weight loss and reduced food and water intake are typical reactions to hypoxia [35,36,37]. Oxygenation of the arterial blood significantly decreased after six days of hypoxia and led to a mainly metabolic acidosis that was associated with an increase in extracellular K^+^ concentration. Under hypoxic conditions, cells switch from oxidative to glycolytic metabolism in order to meet the limited oxygen supply and, even more importantly, to prevent excessive mitochondrial generation of reactive oxygen species (ROS) [38]. Hypoxia-inducible factors (HIFs) trigger this metabolic switch via several mechanisms [39,40,41]. Increased glycolysis is associated with reduced ATP production and increased lactate production, resulting in metabolic acidosis [39,42]. Moreover, a decrease in arterial pO_2_ leads to an increase in plasma K^+^ concentration [43]. Acidosis has a synergistic effect on extracellular K^+^ concentration [44]. A reduced activity of the Na^+^/K^+^ ATPase has been discussed as one of the main mediators of this arterial K^+^ increase [45,46]. Hypoxia exerts a depressant effect on the action potential (AP) of the heart, resulting in a depolarization of the resting membrane potential, shortening of AP duration, decrease in AP amplitude, and a reduced recovery of excitability. These effects are aggravated by elevated plasma K^+^ concentration [47,48] and may further impair the already limited cardiac function in hypoxia.

### 4.2. Effects of Prolonged Hypoxia on the LV Function

After 6 days of hypoxia, the function of the LV appeared to be improved compared with 72 h of hypoxia exposure. LVSP, LV dP/dtmax, MAP, and DAP had re-increased almost to the values of the 72N group. HR and CI, however, further decreased by six days of hypoxia, suggesting that the improved pressure and contractility values may not indicate a real recovery of LV function. This is confirmed by the continued low values of SV and EF. The massive increase in LV edP along with the re-increased LV edV point toward blood congestion in the pulmonary circulation, as can be suggested from the significantly elevated relative lung weight. RVSP was, after six days of hypoxia, significantly higher than in the related normoxic group. At the same time, TPR had risen to approximately 150% compared with the 72H group. According to Frank–Starling’s law, the enlarged end-diastolic filling improves the contractility of the LV and enables the LV to counter the increased arterial resistance. However, the mildly improved contractility cannot fully compensate for the load on the LV from increased filling due to elevated RVSP on the one hand and impeded discharge due to elevated TPR on the other hand. As HR continues to fall, CI is also declining to about 75% of normoxic values. Last but not least, the large number of premature deaths among the hypoxic animals is a clear indication of the severe stress on the LV caused by prolonged hypoxia. We suggest that this hypoxic stress in combination with narcosis caused a decompensation of the LV in some animals, with a subsequent massive backlog into the lungs and into the RV, ultimately leading to the aRVF found at necropsy.

The heart is absolutely dependent on aerobic energy production. Reduced oxygen supply to the myocardium, particularly to the LV, compromises mitochondrial respiration, ATP synthesis, and myocardial energetics [22,23,49,50]. To maintain ATP production, the hypoxic heart switches from the preferred fatty acid oxidation to the less oxygen-consuming glucose oxidation [51,52]. Further adaptations to hypoxia, mainly mediated through hypoxia-inducible factor (HIF)-1α, lead to increased glycolysis. A study in dogs under progressive hypoxemia demonstrated that the reduction in LV oxygen consumption was accompanied by signs of anaerobic metabolism and a decrease in LV contractility [21]. Hypoxia impairs mitochondrial oxidative phosphorylation by modifying the activity of the cytochrome chain. This leads to a reduction in ATP synthesis and an increased formation of ROS, while antioxidant defense systems are decreased [53]. The impaired myocardial oxygenation leads to oxidative and nitrosative stress and further damage to the heart. In a recent study, we demonstrated a significant increase in markers of nitrosative stress, ATP deficiency, and apoptosis in the hearts of hypoxic rats, which were associated with a pronounced depression in the LV contractile function and cardiac output [28]. These findings suggest that oxidative/nitrosative stress, apoptosis, and a reduced myocardial energy state are significant contributors to the LV dysfunction in rats under hypoxic conditions. In addition, both hypoxia and oxidative stress impair pancreatic β-cell function via several molecular mechanisms, resulting in a decreased insulin secretion [54,55]. Moreover, oxidative stress can reduce the expression of the insulin-sensitive glucose transporter GLUT-4, thus reducing cellular glucose uptake [54]. The increased blood glucose concentration observed in the present study indicates that an impaired glucose uptake may have contributed to the poor general condition and LV dysfunction of the hypoxic rats. Finally, acidosis and the increased plasma K^+^ concentration may have affected the excitability of the myocardial cells [48,56] and thus contributed to the LV depression and to the significant bradycardia in the hypoxic rats. However, not only the LV is affected by prolonged hypoxia. The total peripheral resistance, which was almost on a normoxic level after three days of hypoxia, increased significantly after six days of hypoxia exposure. A recent study in humans demonstrated that increased α-adrenergic signaling in chronic hypoxia led to an impaired endothelial-dependent dilation in resistance arteries [57]. This is in line with previous results in rats during short-term exposure to hypoxia. While 24 h of hypoxia alone had no clear effect on TPR, the application of NE increased it, and the α-adrenergic blockade with prazosin decreased TPR [26]. The elevated TPR observed in the 144H group of the present study may provide an explanation for the mild increase in LVSP in this group. In this context, we would assume that the lower LVSP in the 144N group could be a result of the reduced TPR of these animals. The decreased HR and CI of the 144H animals suggest that the pump function of the hypoxic LV is really impaired, and the elevated TPR has contributed to this dysfunction.

### 4.3. Effects of Prolonged Hypoxia on the RV Function and the Lungs

In contrast to the LV, RV function was not reduced under prolonged hypoxia, but even mildly improved. Of note, markers of nitrosative stress, ATP deficiency, and apoptosis were increased in the RV after 72 h of hypoxia to about the same extent as in the LV [28]. We assume that these injuries have almost no measurable impact on RV function, as the work of the RV is about 5–6 times lower than that of the LV. The compromised LV pump function causes a backlog into the pulmonary vascular bed. The RV overcomes this increased afterload by generating a higher systolic pressure, but the compromised ability of the LV to forward the blood from the pulmonary circulation promotes blood congestion in the lungs and formation of a hydrostatic edema.

The pathogenesis of the most typical form of hypoxia-induced PE, the high-altitude PE (HAPE), is based on elevated pulmonary capillary pressure due to an uneven distribution in hypoxic pulmonary vasoconstriction [7]. A mismatch in the pump functions of the RV and LV may further increase the hydrostatic pressure in the pulmonary capillaries. The results of the multiple linear regression showed that hypoxia was the most important predictor of the PE severity, while the cardiac predictors LVSP and RVSP had only a weak influence on the PEI. This result confirms the generally accepted definition of the hypoxia-induced edema (e.g., the HAPE) as a non-cardiogenic edema. People who develop HAPE show an abnormally increased pulmonary arterial pressure after acute or prolonged exposure to hypoxia but no indication of LV failure [7]. However, a previous study on rats infused with norepinephrine showed a significant increase in RVSP and TPR, which was accompanied by elevated pleural transudation. This hemodynamic situation had induced a backlog of blood from the LV into the pulmonary vascular bed, and consequently, increased filtration into the pulmonary interstitium [58]. These findings emphasize that an impaired cardiac function may contribute to the persistence and deterioration of hypoxia-induced PE, even though the impact of hypoxia on the pulmonary vessels has many more facets than only a hemodynamic imbalance between the left and right ventricles of the heart.

Moreover, the formation of HAPE is accompanied by inflammatory processes [59,60], which are considered not to be the cause of HAPE but to maintain and aggravate the edema [61]. Our results confirm this view by demonstrating that PE and the expression of the proinflammatory cytokine TNFα develop at least partly independently of each other. While the PEI remains relatively stable between three and six days of hypoxia, TNFα expression further increases during this period, indicating persistent inflammation. The elevated pulmonary capillary pressure, which is enhanced by the pumping mismatch between the RV and LV, may cause capillary wall stress [10,11]. This is reflected in the high protein concentration of the pleural fluid (PF). Normally, PF is a transudate, which means it contains little protein. A high protein concentration in the PF (more than 50% of the serum protein concentration), as observed in the present study, points toward damage to the capillary walls [33,34]. The concomitant inflammation additionally aggravates the injury, thus impeding resolution of the edema. However, the fluid had not yet entered the alveoli, as confirmed by histology and the low protein concentration in the BAL fluid.

Fluid filtration into the pleural space is an important mechanism for draining excess fluid in the pulmonary interstitium [62,63]. The increased formation of PF might have limited the aggravation of PE. This may explain why the PE was mainly confined to the pulmonary interstitium and did not invade the alveoli, as reflected in the histological image and in the W/D ratios below six in all groups [64]. However, transudation into the pleural space is a limited counter-mechanism, as indicated by the unabated persistence of PE and the significantly increased relative lung weight after six days of hypoxia. The PE is supposed to impede the alveolar gas exchange. Consequently, the metabolic acidosis cannot be effectively compensated for by increased ventilation or may even be aggravated by a respiratory component.

Another counter-mechanism to reduce the fluid load to the lungs under hypoxic conditions is the reduction in plasma volume. A reduced plasma volume means a reduced load for the heart, thus improving oxygen delivery to the tissues and organs. Both humans and rats increase their urine excretion upon exposure to hypoxia [5,6,65]. Moreover, we observed that the hypoxic rats drank significantly less than the normoxic ones. Decreased fluid intake additionally contributed to the reduction in plasma volume, thus counteracting the formation of PE. The animals in the 72H group showed an elevated protein concentration in the serum, indicating a relative hypovolemia. In contrast, the serum protein concentration in animals with 144 h of experiment duration was significantly lower, suggesting that the fluid management in these animals might have been impaired, probably due to the prolonged NaCl infusion. It has been reported that people developing high-altitude diseases presented an increase in total body water accompanied by a reduction in total water loss [66]. In a previous rat study, we had observed that PE and inflammation regressed by seven days of hypoxia exposure. However, these animals did not receive infusion [30]. A relative fluid overload due to prolonged infusion over six days would impede the resolution of PE in these animals and might explain the PE persistence, particularly in the hypoxic ones.

However, the findings of this study do not apply only to persons at high altitude but also to patients suffering from pulmonary diseases associated with generalized hypoxia. Chronic obstructive pulmonary disease (COPD), a disease that affects around 480 million people worldwide [67], is characterized by arterial hypoxemia. Typically, COPD is the leading cause for RV hypertrophy and/or dilation, called cor pulmonale. However, a dysfunction of the RV may occur before pulmonary hypertension becomes evident [68]. Moreover, COPD patients without overt cardiovascular disease can present signs of sub-clinical LV and RV dysfunction, which have arisen as a result of COPD [69,70]. COPD patients may also develop massive pulmonary edema, in particular in periods of acute exacerbation [71], and 6–10% presented an elevated hematocrit as an indication of a secondary polycythemia [72]. Generalized hypoxia, however, may also be a serious condition in acute or sub-chronic pulmonary diseases such as SARS-CoV-2. Out of 195 COVID-19 patients,13% showed severe hypoxemia without a clinical correlation of respiratory symptoms (so-called silent hypoxia) [73]. Those patients can often develop cardiopulmonary complications such as cardiac injury or PE within a relatively short period of time, which can make the treatment of these patients even more difficult [74,75].

### 4.4. Limitations of the Study

Our experimental equipment did not allow for the measurement of urine excretion or for urine analysis, which is a major limitation to this study as it would indicate possible impairments in the fluid and salt balance. Future studies should include metabolic cages allowing determination of the urine excretion rate in addition to the measurement of fluid intake. Those studies might be supplemented by measurements of the hormones involved in the regulation of fluid balance, such as the renin–angiotensin system, which is known to be activated under hypoxia [76]. In addition, hormone activities and renal function parameters could provide further predictors of PE formation.

Further, echocardiographic examinations would provide a more detailed assessment of cardiac function in normoxia and hypoxia as well. In particular, it would allow measuring the LV myocardial twist to detect contractile impairments that cannot be assessed in terms of pressures and volumes in the cardiac ventricles.

The present experiments were performed on female rats only to allow comparison with previous studies by our group [8,30,58]. The estrous cycle has little or no effect on the cardiovascular response to stress in rats [77]. A comparative study on rats revealed no significant differences in blood pressure and heart rate between adult male and female rats both in normoxia and normobaric hypoxia. However, the ventilatory response to hypoxia was higher in female than in male animals [78]. A similar result was found in mice. In addition to an augmented HVR, reduction in HR and activity as well as the development of RV hypertrophy were lower in female than in male mice in chronic hypoxia [79], suggesting that hypoxia-induced damage to the cardiopulmonary system might have been more severe in male rats.

## 5. Conclusions and Future Prospects

The benefit of the present study is the comprehensive investigation of cardiovascular, pulmonary, and blood parameters in a model of sub-chronic hypoxia. Despite detectable signs of hypoxic acclimatization such as increased cHb and Hct, sustained hypoxia induced a further reduction in LV function. Although LV systolic pressure and contractility suggest an apparent improvement, the HR, diastolic parameters such as LV edP, and cardiac output indicate that LV function deteriorated with increasing duration of exposure to hypoxia. As the RV function was not reduced by hypoxia, a pumping mismatch between the ventricles resulted that may have maintained and aggravated PE. The additional inflammation in the lungs may finally have led to capillary wall stress, as suggested by a protein-rich PF. These findings have implications for humans in hypoxic situations, such as travelers to high altitude or patients suffering from generalized hypoxia, suggesting that hypoxia-induced cardiopulmonary malfunction requires a constant monitoring of SaO_2_ and/or paO_2_ and a rapid termination of the hypoxic condition.

Future studies investigating the myocardial injuries induced by hypoxia on a cellular level have to be performed. These analyses should focus on markers of oxidative and nitrosative stress and of impaired energy metabolism as well as on markers of apoptosis in both ventricles of animals exposed to hypoxia over six days. Such damage to the myocardial cells is assumed to be the main reason for the depression of the LV in hypoxia. In addition, similar markers should be detected in the lungs of these animals to explain the hypoxia-induced pulmonary injury in more detail.

## Figures and Tables

**Figure 1 cimb-47-00125-f001:**
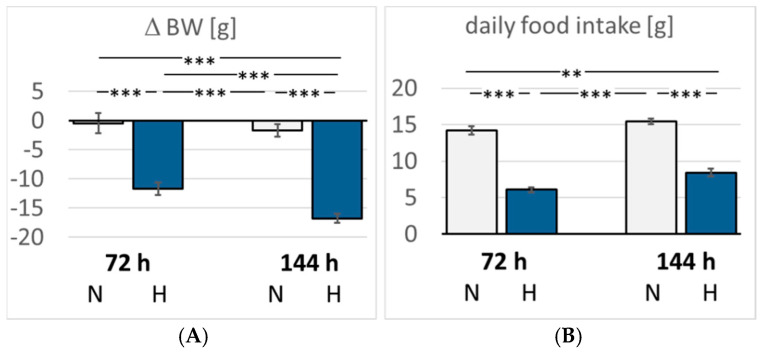
(**A**): Change in body weight (ΔBW) (g); (**B**): Daily food intake (g). Data are given as means *±* SEM. N, normoxic groups; H, hypoxic groups. Significant differences between groups are indicated by asterisks: ** *p* < 0.01; *** *p* < 0.001.

**Figure 2 cimb-47-00125-f002:**
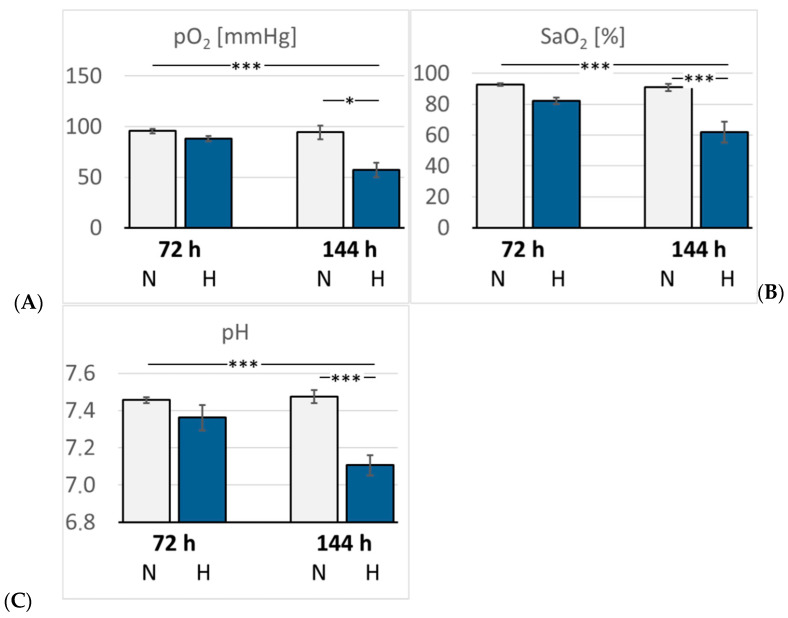
(**A**): Partial pressure of oxygen (pO_2_) (mmHg); (**B**): Arterial oxygen saturation (SaO_2_) (%). (**C**): pH. Data are given as means *±* SEM. N, normoxic groups; H, hypoxic groups. Significant differences between groups are indicated by asterisks: * *p* < 0.05; *** *p* < 0.001.

**Figure 3 cimb-47-00125-f003:**
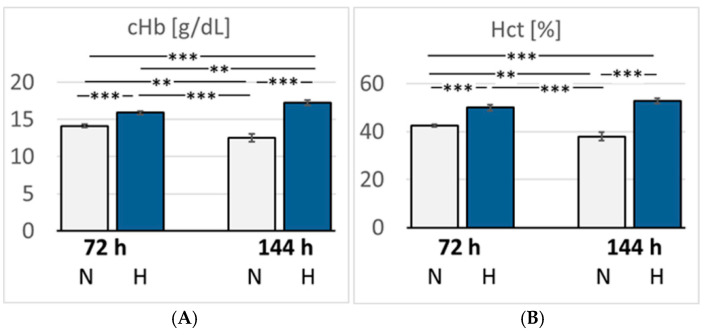
(**A**): Concentration of hemoglobin (cHb) (g/dL); (**B**): Hematocrit (Hct) (%). Data are given as means *±* SEM. N, normoxic groups; H, hypoxic groups. Significant differences between groups are indicated by asterisks: ** *p* < 0.01; *** *p* < 0.001.

**Figure 4 cimb-47-00125-f004:**
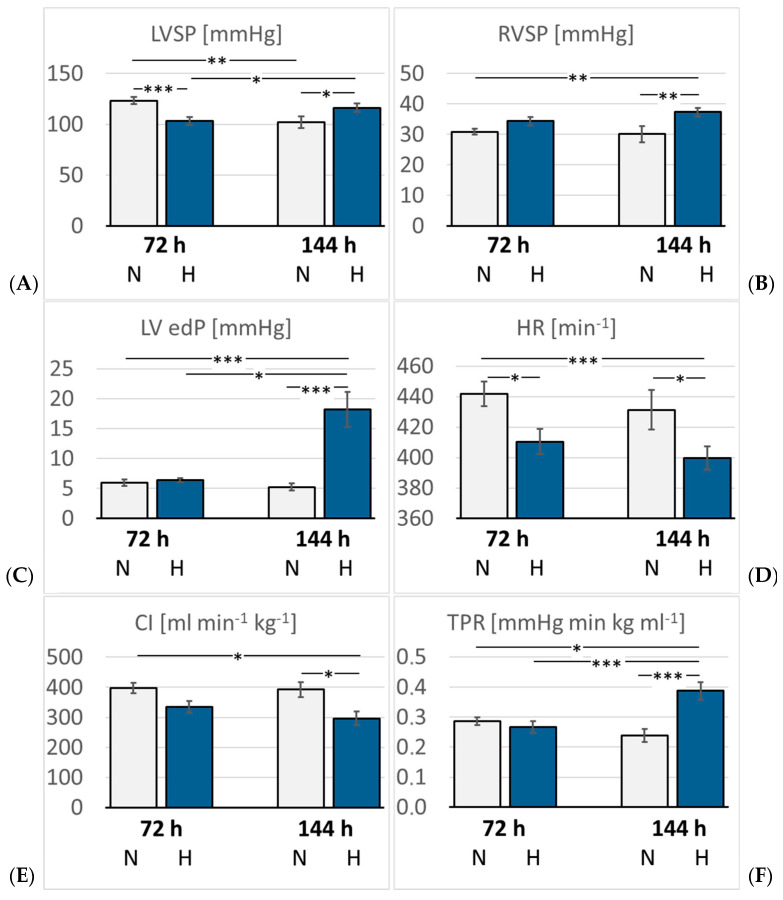
(**A**): Left ventricular systolic pressure (LVSP) (mmHg); (**B**): Right ventricular systolic pressure (RVSP) (mmHg); (**C**): Left ventricular end diastolic pressure (LV edP) (mmHg); (**D**): Heart rate (HR) (min^−1^); (**E**): Cardiac index (CI) (mL × min^−1^ × kg^−1^); (**F**): Total peripheral resistance (TPR) (mmHg × min × kg × mL^−1^). Data are given as means *±* SEM. N, normoxic groups; H, hypoxic groups. Significant differences between groups are indicated by asterisks: * *p* < 0.05; ** *p* < 0.01; *** *p* < 0.001.

**Figure 5 cimb-47-00125-f005:**
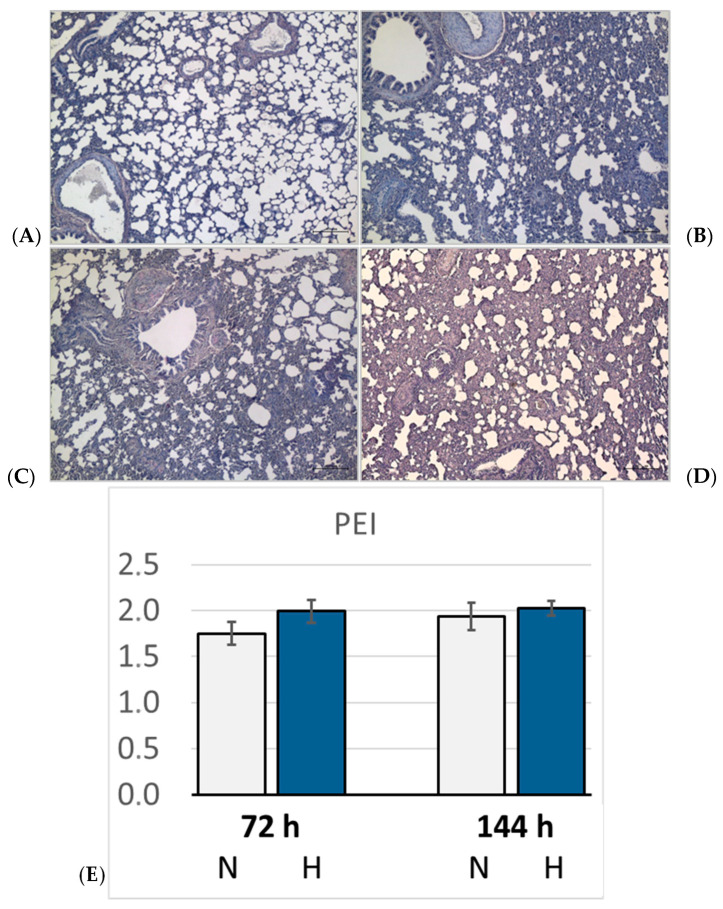
Pulmonary edema. Representative histological images (5× magnification) from: (**A**): 72N; (**B**): 72H; (**C**): 144N; (**D**): 144H; (**E**): Pulmonary edema index (PEI) expressed in arbitrary units. Data are given as means *±* SEM. N, normoxic groups; H, hypoxic groups.

**Figure 6 cimb-47-00125-f006:**
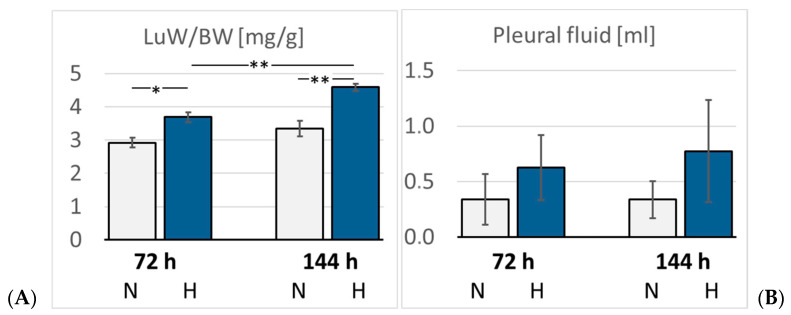
(**A**): Lung weight/body weight (LuW/BW) (mg/g); (**B**): Pleural fluid volume (mL). Data are given as means *±* SEM. N, normoxic groups; H, hypoxic groups. Significant differences between groups are indicated by asterisks: * *p* < 0.05; ** *p* < 0.01.

**Figure 7 cimb-47-00125-f007:**
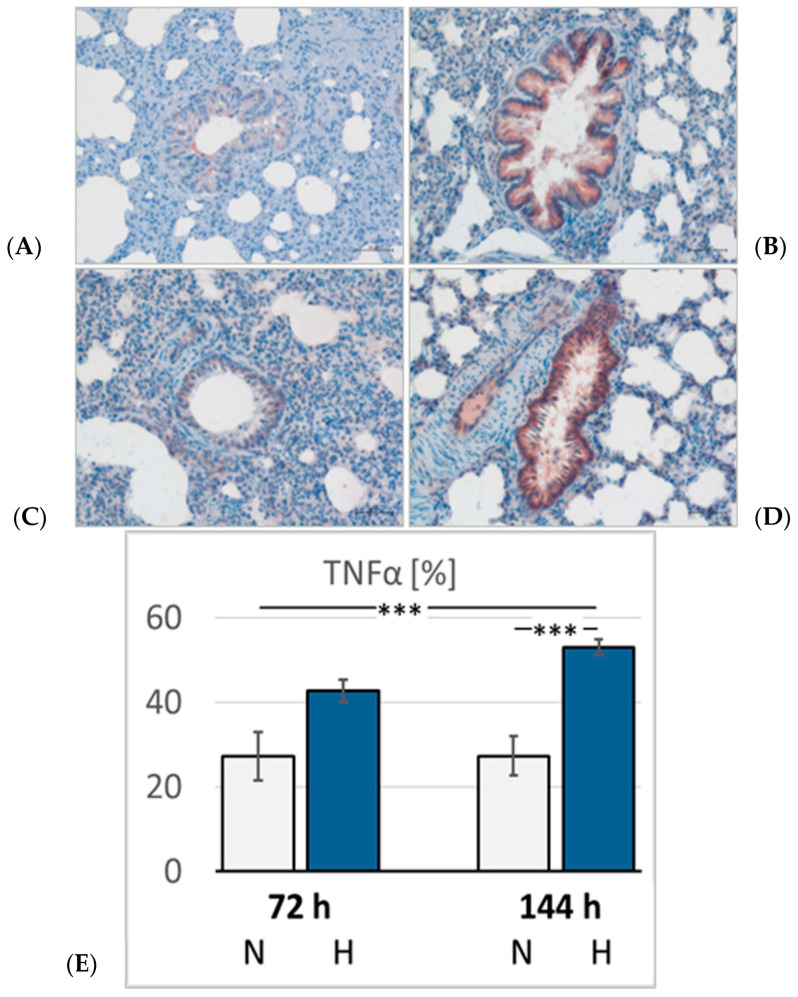
Tumor necrosis factor alpha (TNFα) in the lung. Representative immunohistological images (20× magnification) from: (**A**): 72N; (**B**): 72H; (**C**): 144N; (**D**): 144H. (**E**): Abundance of TNFα in the groups expressed as a percentage of positive area related to the bronchial surface area of the specimen (%). Data are given as means ± SEM. N, normoxic groups; H, hypoxic groups. Significant differences between groups are indicated by asterisks: *** *p* < 0.001.

**Table 1 cimb-47-00125-t001:** Blood analysis data.

	72N	72H	144N	144H
paCO_2_ (mmHg)	38.1 (34.3; 41.2)	33.3 (27.0; 38.3)	38.2 (33.2; 44.0)	40.7 (36.2; 54.0)
Lac (mmol/L)	0.9 (0.3; 1.6)	3.0 (1.7; 5.8)	2.4 (1.2; 3.9)	1.9 (1.5; 6.7)
Glu (mmol/L)	8.6 (7.8; 12.8)	11.2 (8.6; 13.6)	10.2 (10.0; 10.3)	11.8 (10.0; 19.2)
K^+^ (mmol/L)	3.5 (3.3; 3.7)	4.8 (4.3; 5.4) **	3.5 (3.3; 3.8) #	4.1 (3.7; 5.0)
Na^+^ (mmol/L)	132 (126; 136)	142 (139; 146) **	133 (120; 135) #	136 (128; 139)

All values were measured in arterial blood: paCO_2_, partial pressure of CO_2_; Lac, lactate concentration; Glu, glucose concentration; K^+^, K^+^ concentration; Na^+^, Na^+^ concentration. Data are given as median (25th, 75th percentiles). Significance marks: significant vs. time-corresponding N: ** *p* < 0.01; significant vs. 72H: # *p* < 0.05.

**Table 2 cimb-47-00125-t002:** Hemodynamic data and heart weight.

	72N	72H	144N	144H
LV dP/dt max (mmHg/s)	10,419 ± 622	8365 ± 569	9315 ± 992	10,108 ± 609
LV dP/dt min (mmHg/s)	−11,896 ± 468	−10,176 ± 614	−10,570 ± 1091	−10,154 ± 716
DAP (mmHg)	98.4 ± 3.9	79.8 ± 3.4 **	80.9 ± 5.7 +	91.9 ± 3.9 #
MAP (mmHg)	109.6 ± 3.6	91.4 ± 3.6 **	91.4 ± 5.6 +	100.8 ± 5.0
LV edV (µL)	311.0 ± 9.8	279.6 ± 9.7	274.6 ± 21.3	312.0 ± 15.5
SV (µL)	217.2 (203.8; 229.1)	166.2 (158.6; 214.6)	201.3 (193.6; 234.0)	172.0 (141.4; 238.1)
EF (%)	62.8 (59.9; 64.3)	52.0 (43.1; 63.9)	61.2 (57.7; 71.8)	50.6 (41.0; 67.0)
SW (mmHg × µL)	18,512 ± 2174	12,852 ± 1513	15,929 ± 985	15,228 ± 2666
RV dP/dt max (mmHg/s)	2307 ± 183	2458 ± 192	2456 ± 274	2620 ± 160
RV dP/dt min (mmHg/s)	−2098 ± 174	−1928 ± 137	−1986 ± 195	−2341 ± 157
HW/BW (mg/g)	3.20 (3.06; 3.38)	3.41 (3.13; 3.91)	2.78 (2.69; 2.98) ##	3.37 (3.10; 3.58) **

LV/RV dP/dt max, left/right ventricular maximal velocity of increase in pressure (mmHg/s); LV/RV dP/dt min, left/right ventricular maximal velocity of decrease in pressure (mmHg/s); DAP, diastolic aortic pressure (mmHg); MAP, mean aortic pressure (mmHg); LV edV, left ventricular end-diastolic volume (µL); SV, stroke volume (µL); EF, ejection fraction (%); SW, stroke work (mmHg × µL); HW/BW, heart weight/body weight (mg/g). Data are given as mean ± SEM if normally distributed, otherwise as median (25th, 75th percentiles). Significance marks: significant vs. time-corresponding N: ** *p* < 0.01; significant vs. 72H: # *p* < 0.05, ## *p* < 0.01; significant vs. 72N: + *p* < 0.05.

**Table 3 cimb-47-00125-t003:** Protein concentrations.

	72N	72H	144N	144H
[P] S (g/mL)	54.7 (0.47; 0.88)	61.6 (55.8; 76.8)	42.7 (32.5; 50.4) ##	49.2 (40.2; 60.1) #
[P] BALF (g/mL)	0.58 (46.9; 63.8)	0.35 (0.30; 0.54)	0.75 (0.65; 0.84)	0.67 (0.46; 0.82)
[P] PF (g/mL)	18.7 ± 3.8	31.0 ± 4.7	20.5 ± 5.6	28.5 ± 2.8
[P] PF/[P] S	0.29 ± 0.05	0.43 ± 0.06	0.47 ± 0.13	0.63 ± 0.04 ++

[P], protein concentration in: S, serum; BALF, bronchoalveolar lavage fluid; PF pleural fluid. Data are given as mean ± SEM if normally distributed, otherwise as median (25th, 75th percentiles). Significance marks: significant vs. 72H: # *p* < 0.05; ## *p* < 0.01; significant vs. 72N: ++ *p* < 0.01.

## Data Availability

Data are available on request from the corresponding author.

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
