# Peer review of "Effects of a Three-Day vs. Six-Day Exposure to Normobaric Hypoxia on the Cardiopulmonary Function of Rats"

_cimb, 2025, doi:10.3390/cimb47020125_

Round 1

Reviewer 1 Report

Comments and Suggestions for Authors

Reviewer report

Effects of a three-day vs. six-day exposure to normobaric hypoxia on the cardiopulmonary function of rats

This is a well-organized study showing the importance of cardiopulmonary changes during a prolonged but still subchronic exposure to hypoxia. Study shows well explained introduction, background, and appropriate results. There are few sections where manuscript can be improved.

a) Please represent the study in the form of a graphical abstract which will be easy to understand.

b) What will be the effect if the study is carried out in female mice please explain.

c) Authors should have measured the cardiac functions by echocardiography instead of other techniques used in this study.

d) What is the rationale of selecting female SD rats only please justify.

e) Does TNF alpha is the only protein in the bronchial area please list out all the other proteins elevated in the bronchial area and justify why authors have not performed Immunohistochemistry for the same.

f) This article lacks other molecular biology techniques like western blotting and RT-PCR authors should have performed these experiments.

Author Response

We thank this reviewer for his/her valuable comments.

This is a well-organized study showing the importance of cardiopulmonary changes during a prolonged but still subchronic exposure to hypoxia. Study shows well explained introduction, background, and appropriate results. There are few sections where manuscript can be improved.

a) Please represent the study in the form of a graphical abstract which will be easy to understand.

Response: We have created a graphical abstract.

b) What will be the effect if the study is carried out in female mice please explain.

Response:We found many hypoxia studies on mice, which were not well comparable with the present study, often with other exposure times, but most of these studies were performed in male mice. A study comparing the responses to hypoxia in male and female mice revealed that female mice showed a higher hypoxic ventilatory response and a lower cardiac dysfunction than male mice. This finding is in agreement with a comparative study on male and female rats in hypoxia. This was inserted into the Limitations section. (Please see also our response to your comment d).)

c) Authors should have measured the cardiac functions by echocardiography instead of other techniques used in this study.

Response: We stated in the Limitations section that echocardiographic examinations would provide a more detailed assessment of cardiac function. Echocardiography and heart catheterization provide a different set of functional parameters, which complement each other very well. Unfortunately, we had no equipment for echocardiography available in the laboratory where the experiments were performed. An echocardiography device would have been available on a remote site, but could not regularly be moved to the experimental lab. The experimental approach required to have the hypoxic chamber very close to the site of hemodynamic measurements, and this equipment could also not be moved to another place. Prior to the experiments of the present study, we performed some preliminary experiments including echocardiographic examinations in addition to heart catheter examinations. However, the time duration of the total procedure increased remarkably and caused much more stress to the animals. Several animals did not survive this procedure. For our project, the heart pressures and all the parameters obtained by heart catheterization were more important. Therefore, we had to abandon the echocardiographic examinations for this study.

d) What is the rationale of selecting female SD rats only please justify.

Response: The main reason for choosing female rats only was comparability with previous studies. Studies showed that the estrous cycle has little or no effect on the cardiovascular response to stress in rats. While cardiovascular parameters showed no significant differences between male and female rats, the hypoxic ventilatory response was higher in female rats suggesting that we might have found more severe cardiopulmonary damage if we had performed this study on male rats. We inserted the following text into the Limitations section:

“The present experiments were performed on female rats only to allow comparison with previous studies of our group (Rassler et al., 2001, 2003, 2007). The estrous cycle has little or no effect on the cardiovascular response to stress in rats (Sharp et al. 2002). A comparative study on rats revealed no significant differences in blood pressure and heart rate between adult male and female rats both in normoxia and normobaric hypoxia. However, the ventilatory response to hypoxia was higher in female than in male animals (Mortola, Saiki, 1996). A similar result was found in mice. Besides an augmented hypoxic ventilatory response, reduction in heart rate and activity and development of RV hypertrophy were lower in female than in male mice in chronic hypoxia (Wearing, Scott, 2022), suggesting that hypoxia-induced damage to the cardiopulmonary system might have been more severe in male rats.”

e) Does TNF alpha is the only protein in the bronchial area please list out all the other proteins elevated in the bronchial area and justify why authors have not performed Immunohistochemistry for the same.

Response: At present, TNFalpha was the only protein investigated in the lungs by now. In the future, we will analyze more parameters. As we stated in the „Conclusions and future prospects“ section, we are planning further immunohistochemical analyzes both in heart and lung tissue such as markers of oxidative/nitrosative stress and apoptosis. However, these analyses have not been performed yet and will take time until completion.

f) This article lacks other molecular biology techniques like western blotting and RT-PCR authors should have performed these experiments.

Response: In a previous study on rats under hypoxia (Rassler et al., 2007), we have determined the expression of proinflammatory cytokines such as IL-1alpha, IL-1beta, IL-6, and TNFalpha in the lung on mRNA level (using ribo-nuclease protection assay). This analysis did not provide significant differences between normoxic and hypoxic animals. As the immunohistochemical images in the present manuscript show, TNFalpha is strictly localized in the peribronchial regions, and this also holds true for histological signs of inflammation. Due to the regional differences in the expression of inflammation markers, the total expression as determined in analyses of homogenized tissue increases only slightly and not significantly. Western blotting or RT-PCR have the same weakness. Therefore, we preferred immunohistochemistry as this method is able to show local differences in the expression of TNFalpha.

Reviewer 2 Report

Comments and Suggestions for Authors

The article entitled "Effects of a three-day vs. six-day exposure to normobaric hypoxia on the cardiopulmonary function of rats"is interesting as it discuss a novel approach and effect of hypoxic exposure. However, I would have some suggestions:

1. What is the main importance of this exposure? Just for people who leave at high altitude? Could you use this exposure as a therapy in some specific condition? I would comment more about the clinical importance and future perspective of this investigation.

2. Did the authors also investigate other biological markers? Did they have a significant modification?

3. It would be interesting to make a cross analysis between cardiovascular modification and pulmonary modification. Could one of them influence the other?

4. More than this, an analysis between cardiovascular, pulmonary system and biological aspect would be interesting to discuss.

Author Response

We thank the reviewer for his/her valuable comments.

The article entitled "Effects of a three-day vs. six-day exposure to normobaric hypoxia on the cardiopulmonary function of rats"is interesting as it discuss a novel approach and effect of hypoxic exposure. However, I would have some suggestions:

  1. What is the main importance of this exposure? Just for people who leave at high altitude? Could you use this exposure as a therapy in some specific condition? I would comment more about the clinical importance and future perspective of this investigation.

Response: Many hypoxia studies investigate either short-term (1-3 days) or long-term (2-4 weeks) of hypoxia exposure. Here, we were interested in a model of a subchronic exposure to hypoxia. This has been stated in the Introduction section.

The results of this study can be applied to travelers to high altitude who spend several days in this environment. However, it also can be applied to patients suffering from conditions associated with marked hypoxemia. Both subchronic and chronic pulmonary diseases may lead to similar consequences and complications such as those found in our rat model. This has been discussed in more detail (please, see Discussion section, end of subsection 4.3). As these cardiopulmonary injuries directly result from the arterial hypoxemia, monitoring of arterial SO2 and/or pO2 and elimination of the hypoxic condition are of high importance for patients. This has been emphasized in the Conclusions section.

  1. Did the authors also investigate other biological markers? Did they have a significant modification?

Response: We have not investigated other biological markers. As we had stated in the Limitations section, some additional markers, e.g. of renal function might have added more insights. We have added some text into this section. Further markers of oxidative/nitrosative stress, impaired energy metabolism and apoptosis both in the heart and the lungs would be very interesting. Their analysis is planned in the near future but has not been performed yet. We had stated this in the “Conclusions and future prospects” section.         

  1. It would be interesting to make a cross analysis between cardiovascular modification and pulmonary modification. Could one of them influence the other?

Response: We have performed a multiple linear regression with LVSP, RVSP, and hypoxia as predictors of the criterion PEI. The method has been described in the Methods section (subsection 2.9. Statistical analysis). The results have been inserted as a new subchapter 3.5. Effects of cardiac dysfunction on pulmonary edema. The analysis revealed weak effects of LVSP and RVSP, but a significant effect of hypoxia on PEI.

  1. More than this, an analysis between cardiovascular, pulmonary system and biological aspect would be interesting to discuss.

Response: We have discussed the results of the multiple linear regression in the Discussion section (subsection 4.3. chp. 4.3., Effects of prolonged hypoxia on the RV function and the lungs; L 602-616). On the one hand, this result is in accordance with the generally accepted definition of the hypoxia-induced edema as a non-cardiogenic edema. On the other hand, it demonstrates that an impaired cardiac function may contribute to the persistence and deterioration of hypoxia-induced PE. However, investigation of the renal function and fluid regulation would be a valuable supplement.

Reviewer 3 Report

Comments and Suggestions for Authors

Major:

This study investigates the impact of prolonged normobaric hypoxia on cardiopulmonary function, and the study design is clear and the manuscript is well-organized. However, a major concern lies in the novelty and depth of the manuscript. The use of hypoxia-induced rat models is well-established in previous studies, particularly in the field of pulmonary hypertension. Therefore, it is crucial for authors to emphasize the originality of their work and clearly differentiate it from existing literature to demonstrate its unique contribution.

Minor:

Table 1 should include blood HCO₃- levels to provide a clearer indication of the acidosis status.  Also, incorporating a pressure-volume study to calculate Ees/Ea, if possible, would offer valuable insights into the impact of hypoxia on RV–PA coupling. 

Author Response

We thank this reviewer for his/her valuable comments.

Major:

This study investigates the impact of prolonged normobaric hypoxia on cardiopulmonary function, and the study design is clear and the manuscript is well-organized. However, a major concern lies in the novelty and depth of the manuscript. The use of hypoxia-induced rat models is well-established in previous studies, particularly in the field of pulmonary hypertension. Therefore, it is crucial for authors to emphasize the originality of their work and clearly differentiate it from existing literature to demonstrate its unique contribution.        

Response: There are many hypoxic rat models, however, most of them either investigate short-term (acute and subacute model) or long-term (chronic model, > 2 weeks) exposure to hypoxia. A subchronic exposure has only rarely been investigated. Moreover, most of the previous studies either examined cardiac function or pulmonary injury. To study these two interrelated function systems was of particular interest to us. We have stated this in the Introduction and in the Conclusion section:

Introduction: “Most animal studies on the effects of hypoxia have applied either short-term (1-3 days) or long-term (2-4 weeks) of hypoxia exposure. As we were interested to investigate an intermediate time-interval, we compared the effects of three-day and six-day exposure to hypoxia on heart and lungs of rats. (L102-106) … Many other studies on hypoxia explored either cardiac function or pulmonary function, but there are only very few studies that investigated the effects on both systems and their possible interrelations. With its experimental approach, the present study might contribute to close this gap, thus deepening the insight into the complex systemic effects of generalized hypoxia.” (L113-117)

Conclusion: “The benefit of the present study is the comprehensive investigation of cardiovascular, pulmonary, and blood parameters in a model of subchronic hypoxia.” (L702-703)

 In addition, we have performed a multiple linear regression with LVSP, RVSP, and hypoxia as predictors of the criterion PEI. This analysis emphasizes the close interrelationship between cardiovascular and pulmonary reactions to hypoxia.

Minor:

Table 1 should include blood HCO₃- levels to provide a clearer indication of the acidosis status.  Also, incorporating a pressure-volume study to calculate Ees/Ea, if possible, would offer valuable insights into the impact of hypoxia on RV–PA coupling. 

Response:  Our blood gas analyzer ABL800 BASIC was not able to measure HCO₃- in the blood, so we had to rely on pH, pCO2, and lactate concentration.

The p-V measurements were performed in the LV only. For measurements in the RV, we only had a pressure transducer. To insert a catheter into the RV, it must be bent at an obtuse angle, but a bent p-V catheter was not available to us.

Round 2

Reviewer 2 Report

Comments and Suggestions for Authors

The authors responded to all of my questions. The article is suitable for publication.

Reviewer 3 Report

Comments and Suggestions for Authors

No additional comments from my side. The current version looks good to me.